# Nanomaterial Shape Influence on Cell Behavior

**DOI:** 10.3390/ijms22105266

**Published:** 2021-05-17

**Authors:** Daniil V. Kladko, Aleksandra S. Falchevskaya, Nikita S. Serov, Artur Y. Prilepskii

**Affiliations:** International Institute “Solution Chemistry of Advanced Materials and Technologies”, ITMO University, 191002 Saint Petersburg, Russia; kladko@scamt-itmo.ru (D.V.K.); falchevskaya@scamt-itmo.ru (A.S.F.); serov@scamt-itmo.ru (N.S.S.)

**Keywords:** nanoparticle, shape, microbial cell, mammalian cell, crystal growth, anisotropic

## Abstract

Nanomaterials are proven to affect the biological activity of mammalian and microbial cells profoundly. Despite this fact, only surface chemistry, charge, and area are often linked to these phenomena. Moreover, most attention in this field is directed exclusively at nanomaterial cytotoxicity. At the same time, there is a large body of studies showing the influence of nanomaterials on cellular metabolism, proliferation, differentiation, reprogramming, gene transfer, and many other processes. Furthermore, it has been revealed that in all these cases, the shape of the nanomaterial plays a crucial role. In this paper, the mechanisms of nanomaterials shape control, approaches toward its synthesis, and the influence of nanomaterial shape on various biological activities of mammalian and microbial cells, such as proliferation, differentiation, and metabolism, as well as the prospects of this emerging field, are reviewed.

## 1. Introduction

Nanomaterials are widely used in medicine as platforms for developing advanced drug delivery systems with controllable drug loading efficacy, biodistribution and cell/tissue targeting, therapeutic actions, cytotoxicity, selectivity, imaging ability, blood circulation time, half-life, and excretion. It is widely thought that all these properties are connected only with nanomaterial surface chemistry, total surface area, hydrodynamic size, the loaded drug, etc. The phenomenon of nanomaterial shape is usually considered in the context of systemic toxicity, biodistribution, and blood circulation time. However, its influence on the biological activities of mammalian and microbial cells has been reported in many articles, such as shape-induced directed differentiation [1], cellular death via apoptosis [2], necrosis [3], gene transfection and transfer [4], metabolism alteration [5], and other processes. These effects arise from different surface areas, uptake level, protein corona, physical disruption of the cell membrane, and particles’ wettability and surface curvature [6]. Moreover, the shape control of particle morphology demonstrates the importance of the ratio of the different crystal facet surface areas, which have a different dissolution rate in solution [7].

The shape control of nanomaterials represents a major field in materials science, and a large number of approaches exist towards the production of sphere- [8], ellipsoid- [9], dumbbell- [10], cube- [11], polyhedra- [12], rod- [13], urchin- [14], star- [15], chain- [16], ribbon- [17], hollow [18], prism- [19], and hexagon-shaped [20] nanomaterials. Nanomaterial shape control stems from effects such as selective adsorption growth of reactive facets; spontaneous aggregation and agglomeration; seeded growth on particularly-shaped templates; controllable crystal fusion via orientation attachment; self-assembly via selective strong interactions, e.g., chemical and hydrogen bonds; and Ostwald ripening directed at free surface energy minimization. Several synthetic approaches exist toward synthesizing nanomaterials with controlled morphologies, e.g., using polymeric additives, surface-capping surfactants, solvo/hydrothermal approaches, etc.

The discussion of particles’ shape in nanomaterials-related biological applications within the comprehensive reviews are usually fragmentary [6,21] and traditionally cover the shape-related toxicity without studying the harmless metabolic alteration, gene expression, or differentiation. Furthermore, review articles usually describe nanoparticles’ effects on mammalian cells without paying sufficient attention to microbial cells [22].

In this review, we focus on the mechanisms involved in nanomaterial shape control, synthetic approaches toward its production, and the influence of nanomaterial shape on various biological activities of mammalian and microbial cells, e.g., proliferation, differentiation, and metabolism, as well as reviewing the prospects of this emerging field.

## 2. Main Mechanisms of Nanoparticle Shape Control

Several physicochemical phenomena can result in the controlled growth of either complex isotropic or anisotropic structures. Synthesis procedures can be controlled and fine-tuned to make use of these effects to obtain nanomaterials of desired shapes and surface morphologies for further particular applications. The final material shape depends strongly on the crystal facets’ surface chemistry, reagents’ diffusion rates, concentrations, convection currents, nucleation type (whether homogenous or heterogeneous), etc. Aggregation, agglomeration, fusion, and self-assembly effects are often observed, which can be driven by weak Van der Waals, hydrophobic, strong Coulomb, and hydrogen bonding interactions. In this section, we provide an overview of the impact of these phenomena on shape-controlled crystal growth. The main mechanisms are presented in Figure 1.

### 2.1. Adsorption Growth (Selective Faceted Growth)

It is well known that the growth rate of NPs is highly dependent on the surface free energy and the number of surface irregularities, such as steps and kinks. NPs with higher free energy and larger amounts of kinks on their surface will grow faster to reduce the surface area. Kinks, being point defects on one-dimensional steps, facilitate rapid facet growth. Their further growth does not require any extra energy, which explains this behaviour, notwithstanding the fact that kinks are not favorable since they cost energy. If sufficient energy is provided for the reaction, crystal nuclei grow under thermodynamic control, which generates mainly isotropic NPs. Particular crystal facets can be blocked with additives, e.g., small molecules, polymers, and surfactants. By lowering facets’ surface tension, selective faceted growth can be achieved, leading to anisotropic growth of the nuclei and subsequent formation of non-spherical, anisotropic NPs. For example, Ha et al. have shown that the shape control of gold NPs can be achieved by simply tuning the concentration and the type of halide ions in the solution, which is due to specific adsorption of these ions onto the {111} plane [23]. At the same time, preferential adsorption of ionic surfactants onto {100} crystal planes leads to an anisotropic growth of nanorods and nanowires. Thus, the initial nucleus shape and the chemistry of its facets strongly influence the final nanocrystal shape, sometimes leading to the formation of anisotropic NPs. Plane twinning in the initial nuclei, introduced, for example, by adding competing stabilizing agents with different binding strengths [24], also reduces the overall symmetry of the crystal, thus inducing the formation of non-spherical NPs. Therefore, specific adsorption of molecules onto the nuclei crystal planes allows one to alter its growth rate and introduce lattice distortions, changing the final crystal shape.

### 2.2. Surfactant Method

Specific adsorption of several surfactants onto the crystal planes often results in the anisotropic growth of mainly 1D morphologies [25]. Surfactants form wormlike micelles in the solution or liquid crystalline phase due to the presence of sterically bulky headgroups in the structure of the surfactant molecules. If the surfactant is zwitterionic or nonionic, a denser bilayer is formed on the nuclei surfaces, hindering ion diffusion and subsequent crystal growth [26]. When two or more surfactants are used together, mixed micelle and subsequent vesicle formation can be achieved [27]. Such vesicles usually have a smaller size and greater hydrophobicities and aggregation numbers due to charge neutralization in the headgroup Stern layer (if surfactants are oppositely charged). Such a compact layer on the particular crystal planes provides almost complete surface passivation, thereby giving no possibility of adatom crystal growth. At the same time, a decrease in the relative viscosity of surfactant mixtures comprising smaller micelles with smaller micelle aggregation numbers facilitates the diffusion processes, thereby supporting an ordered crystal growth. Moreover, such micelles demonstrate greater surface adsorption ability and surface passivation closer to the sizes of the growing nucleation centers. In addition, surfactant hydrophobicity correlating with its tail length was shown to affect the aspect ratio of NPs [28]. Longer hydrocarbon chains form more compact bilayers, thereby creating no possibility of lateral growth and producing rod-shaped NPs. Thus, this approach is very suitable for precise shape control.

### 2.3. Aggregation and Agglomeration

NPs tend to agglomerate and aggregate after collision during Brownian/flow motion in the solution. NPs usually have high surface free energy, and this process minimizes the surface boundary between NPs and the solvent and the corresponding surface free energy [29]. For decades, aggregation was thought to be a negative effect, which was circumvented by making more stable sols using surface coating and surface charge tuning. However, several pieces of research show that this phenomenon can be used to obtain nanomaterials with controllable morphologies by tuning interparticle interactions through the inclusion of repulsive forces to compromise attractive ones [30]. For instance, Ming et al. have demonstrated that large Prussian blue (PB) crystals of cubic shape can be formed by NP aggregation and fusion, accompanied by a transition from porous mesocrystals to dense quasi-single crystals [31]. Although aggregation is mainly caused by weak Van der Waals interactions, agglomeration (either Brownian or gravitational) occurs with the help of chemical bonding between NP surface groups, thus being irreversible, which has a profound effect on its catalytic [32], biological [33], and other activities. In several works, it was demonstrated that agglomeration processes are highly pH- and ionic strength-dependent. Agglomerate size increases with ionic strength and pH values, approaching the isoelectric point (IEP) [34], where NP size also has a profound effect [35].

Moreover, reversible magnetic agglomeration can be utilized. Usually, this process leads to the formation of isotropic nanomaterials, and it is difficult to achieve precise shape control. However, NPs with complex surface morphologies consisting of plates and spikes can be obtained. For instance, Shen et al. obtained spherical structures with complex morphology via the controlled agglomeration of nanorods and nanoplates [36]. The process involved tuning the poly(vinylpyrrolidone) (PVP) concentrations, used as a stabilizer and ionic capping agent. Facet-specific chemical modifications of crystal surfaces can potentially ensure further oriented agglomeration resulting in the formation of anisotropic structures.

### 2.4. Seeded Growth

Heterogeneous nucleation of the target crystals on the surface of pre-nucleated seeds represents another approach towards obtaining anisotropic NPs. The crystallinity of the seeds does not always dictate the crystallinity of the final NPs. For instance, when the seeds are not large enough to preserve their shape due to low free surface energy, they can undergo structural fluctuations, resulting in a crystallinity change [37]. If the lattices of the two materials have a very high mismatch, crystal overgrowth around the initial seed is inhibited. This results in both seed and secondary crystal exposure, which always results in anisotropic structures.

In contrast, when the lattice mismatch is low enough, heteroepitaxial crystal overgrowth occurs. Crystal seed provides a well-defined surface for the overgrowth and also dictates the final shape of the core-shell structures formed [38]. Moreover, when coupled with galvanic replacement, this technique allows one to achieve shape control of hollow NPs from the core-shell structures [39]. Applying this approach and the surfactant method comprising crystal growth within the wormlike micelles, nanorods of high aspect ratios and even nanowires can be obtained [40,41].

### 2.5. Orientation Attachment

Sometimes collision between NPs leads to crystal fusion followed by the formation of a common crystallographic orientation. Usually, this attachment occurs between high-energy facets to reduce the free surface energy, either of pre-aligned nanocrystals or after rotation of misaligned NPs. For example, Zhu et al. demonstrated that after rod-like monocrystal nucleation and growth, followed by their aggregation, the oriented attachment of these aggregates led to the formation of urchin-like NPs [42]. Moreover, if the reaction continued, urchin-like NPs transformed into rod-like aggregates due to crystal fusion, which happens to minimize the free surface energy of the crystal. Schlieche et al. have demonstrated that chlorine-containing solvent-induced oriented attachment of PbS NPs can be utilized to produce two-dimensional materials [43]. The solvent plays an important role in the nucleation and growth of NPs which have reactive {110} facets exposed. Therefore, orientation attachment allows us to obtain large crystals and synthesize complex structures, although in a less predictable manner due to the high complexity of this process.

### 2.6. Self-Assembly

Self-assembly is a complex process mediated by the combination of weak (Van der Waals, hydrophobic interactions) and strong (electrostatic, hydrogen, and coordination bonds) forces, which often results in the formation of hierarchical structures [44]. In the case of the self-assembly of nanoparticles, their shape [45] and size, monodispersity, and surface chemistry, e.g., zeta-potential and surface groups, strongly influence the final morphology of these structures. Since one of the first reports of coupled synthesis and NP self-assembly resulted in the formation of hierarchical structures with controlled organization [46], many research articles have shown the versatility of this approach, namely, the synthesis of LaFeO_3_ perovskite microspheres [47], self-assembly via NP hydrophobicity/hydrophilicity tuning [48], magnetic field-mediated self-assembly [49], self-assembly of highly anisotropic structures [50], etc.

Moreover, programmable self-assembly is rapidly coming into practice, including regioselective isotropic [51] and anisotropic [52] NP surface encoding via modification with DNA oligonucleotides. Hence, the process of self-assembly can be utilized to form complex suprastructures in a highly controllable manner through chemical bonding and physical stimuli, in contrast to aggregation and agglomeration.

### 2.7. Ostwald Ripening

Ostwald ripening is the phenomenon of NP dissolution and reprecipitation driven by the chemical potential, leading to the formation of bigger NPs while their amount decreases. Although this process usually results in isotropic structures, several experimental research articles show the Ostwald ripening-driven formation of anisotropic NPs. For instance, Gao et al. reported the formation of hollow bundle-shaped NaYF_4_ microparticles, where multi-stage shape evolution occurred due to Ostwald ripening [53]. Furthermore, Knusel et al. showed that two-dimensional materials with tunable thickness could be produced due to this effect [54]. Zhang and Wang have also demonstrated that complex shell-in-shell morphologies can be obtained through this process [55]. Therefore, Ostwald ripening can be exploited for the production of hollow—including multi-shell—morphologies and materials with controlled dimensionality.

## 3. Methods of Nanoparticle Shape Control

There are two main approaches to NP synthesis: bottom-up and top-down. The latter is based on physical phenomena, leaving little room for controlling the shape of the nanoparticles. On the other hand, bottom-up approaches are based on solution chemistry and open up more variable parameters to control both size and shape (Figure 2). According to the classical scheme of the formation of nanoparticles in solution, after nucleation, which in most cases determines the size, the parameters of crystal growth determine the morphology of the resulting nanoparticles. There are two models for describing the growth of crystals on the surface of the nuclei. They are surface reactions and monomer diffusion to the particle surface. The concentration of the precursor controls these processes. At a high concentration of the precursor, the growth rate is controlled by diffusion-mediated mechanisms, whereas at a low concentration of the precursor, reaction-limited growth occurs.

Diffusion-mediated growth controls the monodispersity of NPs and the surface reaction-limited growth determines the particles’ final shape. By creating diffusion barriers on the particle surface with surfactants, organic ligands, or stabilizers, it is possible to control both the size distribution and the shape of the final nanoparticles [56].

### 3.1. Concentration of the Precursors

Supersaturation is the main process controlling nucleus formation and its growth rate. When the concentration of the precursor monomer reaches a critical point, spontaneous nucleation occurs, and the concentration of the precursor decreases. The growth of the nuclei is also accompanied by a decrease in the concentration of the precursor during the diffusion-limited growth process. Furthermore, any mechanism described above may occur on the nuclei (see Section 2). Thermodynamically controlled growth results in uniform isotropic particles, whereas kinetic regimes produce anisotropic structures [57]. Thus, the concentration of the precursors is a basic variable parameter that can strongly influence the shape of the final nanoparticles. However, the concentration of the precursor (monomer) can be influenced directly or indirectly. The solvent type, the acidity of the solution, and various complexing agents capable of coordinating with the precursor can affect the mechanism of the process.

A comprehensive review [56] discusses the precursor concentration issue. In the corresponding example [12], Eguchi et al. showed that shape could be controlled only by different amounts of ascorbic acid during the seed-mediated growth of polyhedral gold nanoparticles. This dependence is explained in terms of free planes of gold nuclei. At a high concentration of ascorbic acid, the number of seeds increased rapidly, leading to thermodynamically unstable structures. The number of gold atoms decreased at a lower concentration, giving freedom to the thermodynamic growth method of obtaining nanoparticles. Lai et al. recently reported an elegant way to control the crystallization process (1D to 3D structures) by balancing the precipitate electrolytic dissociation (α) of the reactants and the supersaturation (S) of the solutions. This route is based on the fact that strong electrolytes (α = 1) can produce a maximum number of ions, facilitating the nucleation process. Weak electrolytes release ions gradually, controlling the kinetics of the reaction, which is favorable for the anisotropic growth. It is possible to obtain products with different solubility constants by controlling the values of supersaturations with a precipitating agent which has a different dissociation constant. Thus, when barium fluoride was obtained using a strong precipitant, a supersaturated solution was formed quickly, providing 3D growth of polyhedral structures.

In contrast, using NH_4_HF_2_ instead of NaF leads to anisotropic 1D rod-like structures [58]. A weak NH4HF2/F-1D or 2D structure can be obtained, depending on the level of supersaturation (higher for the formation of 2D structures). A wide range of materials has been prepared using this strategy, including metal oxides, hydroxides, carbonates, molybdates, oxalates, phosphates, fluorides, and iodate with various morphologies. These prove that the α values of reactants and the S values of solutions play key roles in shape control.

Considering the role of the reducing agent, it is worth discussing silver nanoparticles. They have been extensively studied from the size and shape control point of view. Reducing agents are mainly represented by the following substances: sodium borohydride (NaBH_4_), sodium citrate, and polyols (ethylene glycol). Wiley and colleagues synthesized various Ag NP shapes via polyol synthesis, finely controlling the shape based on the kind of dopants added [59]. They achieved control over the seed’s crystallinity and crystal growth selectivity. Adding different dopants during crystal growth, they selectively produced pentagonal nanowires, cuboctahedral, nanocubes, nanobars, bi-pyramids, and nanobeams of silver. Freshly formed nuclei mainly possess twin boundary defects, enabling low surface energy. During growth, they are transformed into multiple twinned, singly twinned, or single-crystal seeds. Those seeds dictate the shapes of the resulting nanoparticles. Single-crystal nanocubes and cuboctahedrons were synthesized during the polyol procedure in the presence of Cl^−^ ions and air. Seeds with twin defects undergo etching, forming single-crystal nanocubes or cuboctahedrons. The final shape can be controlled by varying the reaction time. After 46 h, crystals become 80-nm cuboctahedrons, whereas nanocubes with a 25-nm edge form after 24 h reaction time. Multiply twinned pentagonal nanowires were obtained when Fe(acac)_3_ chloride ions were added during polyol synthesis. If bromide ions were added instead of chloride ions, both nanocubes and nanobars could be obtained. In the same way, but by reducing the number of bromine ions twice, one could obtain silver nanoparticles in the form of bipyramids. Slowing down the reduction rate by decreasing the temperature, but increasing the concentration of silver nitrate and PVP, it was possible to obtain silver nanobeams. Pentagonal nanowires were synthesized in the same study, and it is believed that it is because of PVP’s shape orientation properties.

Many options for obtaining nanoparticles of different shapes described above apply to palladium nanoparticles as well. Xia et al. described several examples in their review [60]. For instance, palladium nanocubes can be obtained by reducing the corresponding salt using PVP as a capping agent and Fe (III) as an etching agent [61].

One more way to achieve anisotropic palladium nanostructures is to control reaction kinetics. Xia et al. prepared Pd nanobars and nanorods [62]. The reaction mixture consisted of KBr, PVP, ethylene glycol, and the Na_2_PdCl_4_ salt as palladium precursor. PVP is considered a relatively mild reducing agent, whereas ethylene glycol is considered more robust. Thus, the reduction rate increased with increasing EG concentration in the reaction mixture, providing kinetic control of the reduction.

The first mention of palladium polyhedra was made by Niu et al. [63]. Polyhedral nanocrystals were prepared from initial nanocubes in solution containing cetyltrimethylammonium bromide (CTAB), H_2_PdCl_4_, ascorbic acid, and KI. Manipulating KI concentrations and reaction temperature, several polyhedral shapes, such as rhombic dodecahedral, cubic, and octahedra, were obtained.

Millstone et al. examined the effect of the concentration of the reducing agent on the shape of gold nanoparticles [64]. Thus, in the reaction mixture, which consisted of the gold precursor (HauCl_4_), surfactant CTAB and ascorbic acid, different amounts of reductant (NaBH_4_) were added. A critical step in achieving anisotropic nanoparticle growth was modulating the ratio of NaBH_4_ to HauCl_4_ to limit the amount of gold consumed during nucleation, thereby influencing the concentration remaining for growth.

In addition to nanoparticles of noble metals, nanoparticles of transition metals and their oxides (magnetite, copper oxide, zinc oxide, etc.) are widely used in biomedical applications. Fatima et al. demonstrated that the shape of the magnetite nanoparticles could be controlled by the type and concentrations of the precursors and capping agents [65]. FeSO_4_ coupled with KOH as a capping agent resulted in cubes and octahedra nanoparticles, whereas FeCl_3_ with ammonium acetate mixture lead to spheres.

### 3.2. Solvents

Since solvents with different functional groups may coordinate with any reactant, this potentially can control the reaction rate. The polyol process is widely used in shape-controlled nanoparticle synthesis. Biacchi et al. demonstrate that the selection of a proper polyol solvent can provide various morphologies of metal nanoparticles [66]. They provide a comprehensive study on polyol solvent influence on Rh nanoparticle shape and size, obtained from precursors with only different anions. Nanoscale Rh icosahedra, cubes, and triangular plates were synthesized. Since different polyol solvents (ethylene glycol, diethylene glycol, triethylene glycol, and tetraethylene glycol) possess different oxidation potentials, it defines particles’ formation temperature, therefore guiding kinetic/thermodynamic control over particles growth [66].

In a review [67], deep eutectic solvents (DES) described as shape-directing agents. It is assumed that halide ions in DES provide shape-directing properties as they do in classical water-assisted synthesis. Moreover, hydrogen bond donors in DES (amine, hydroxyl, and carboxyl groups) exhibit facet-specific adsorption, which results in anisotropic growth. Pioneering work in the shape-controlled synthesis of gold NPs was performed by Sun et al., who achieved the synthesis of shapes such as stars, snowflakes, and thorns [15].

### 3.3. pH

Controlled overgrowth of already synthesized nanoparticles provides an alternate method for tuning nanoparticles’ geometry. Thambi et al. used pre-synthesized gold nanoparticles for further pH-controlled growth in the presence of silver ions. This way, water chestnut, dog bone, nanobar, icosahedron, and octahedron shapes can be formed.

At pH higher than four, kinetically controlled geometries prevailed due to the fast deposition rate. Authors investigate the influence of four pH values: 2, 4, 6, and 8. At the lowest pH (=2), the shape remained unchanged. At higher pH, the deposition rate is much higher, resulting in kinetic growth along the (111) plane instead of (100) and (110)—the lowest energy surface in the presence of CTAB [68]. The Peng group investigated kinetically controlled growth [69]. Gold nanoflowers were synthesized via a pH-dependent seed-mediated approach. Firstly, HauCl_4_ was rapidly reduced by hydroxylamine, followed by the formation of 3-nm nanoparticles. Then, small particles attach to the seed surface resulting in the growth of nanoflowers. With higher pH, more suppressed ripening occurs, providing longer branches for the flowers. Wang et al. developed a methodology for the synthesis of star-shaped nanoparticles. These morphologies can be fine-tuned systematically by simply adjusting the pH of the reaction system with NH_2_OH_3×_HCl as a reductant [70].

There is a way to obtain silver nanowires by merely controlling the pH of the solution without adding surfactants or polymers. Caswell et al. reduced a silver precursor with sodium citrate, using NaOH for pH control, and found that the hydroxyl group concentration was a key factor for production of these nanowires [71].

### 3.4. Polymers and Surfactants

In most cases, the addition of surfactants is inevitable, as the nanoparticles in the solution aggregate due to the high surface energy, which leads to the appearance of large irregular structures. Surfactants reduce NPs’ surface energy via absorption on the edges of crystal nuclei, thereby preventing them from sticking together. Due to this effect, they exhibit shape-directing properties. Surfactants play the role of capping agents during nanoparticle synthesis. By selecting surfactants with different functional groups, it is possible to achieve NP stability and selective growth [56,72]. CTAB and CTAC (cetyltrimethylammonium chloride) have been widely used in the synthesis of noble metals NPs, as mentioned above. Even the length of the hydrophobic chain of CTAB influences the morphology of gold nanostructures [73]. The effect of the surfactant anion is also worth mentioning. With the use of CTAC instead of CTAB, all other things being equal, the particle shape shifts from polyhedral to cubic [72]. Polymer stabilizers are also widely used in the control of the size and shape of nanoparticles, namely, PVP, PAA (poly(acrylic acid), PVA (poly(vinyl alcohol), and others are used [56]. Kedia et al. [74] conducted a systematic study on the synthesis of gold nanoparticles using PVP as a surfactant in various solvents. Their work illustrated cooperative PVP−solvent−metal ion interactions, which led to more/less complex structures.

Generally speaking, most often, anisotropic particles are a consequence of the action of many factors, and not just one. By altering the kinetic control, surfactants, the acidity of the medium, and supersaturation, various forms can be achieved.

## 4. Biological Effect and Application

### 4.1. Mammalian Cells

#### 4.1.1. Cell Death

##### Apoptosis

Using nanoparticles to kill cells is one of the most common tasks. Nanoparticles are usually considered for use as drug carriers and heat or light conductors. However, even in the case of drug delivery and retention, the shape is of great importance [75]. Nevertheless, nanoparticles themselves can cause cell death, and not because of their toxicity. The following are studies showing the difference in the cellular response due to the various nanoparticle shapes.

Gold nanoparticles were among the first to be used in medicine. Widely developed methods for controlling the shape of gold nanoparticles have made it possible to obtain various shapes and sizes. However, when considering gold nanoparticles, it is challenging to separate shape effects from the size or surface functionalization effects. Thus, one of the most common methods of shape control is using surfactants. In [2], gold spheres, rods, and stars were compared. Anisotropic rod and star nanoparticles were obtained using CTAB, which is known for its cytotoxicity. The authors recorded a significant increase in the level of proapoptotic protein (Bax) and a decrease in that of anti-apoptotic protein (Bcl-2) for star and rod shapes. The authors also recorded the induction of autophagy in tested cells. In general, osteosarcoma MG-63 and 143B cells were more susceptible to nanoparticles than normal osteoblast hFOB 1.19 cells.

In most cases, increased toxicity (and subsequently, induction of cell death) is associated with “sharp” shapes [76]. However, some authors report opposite results. For instance, in a work by Favi et al., commercially purchased citric-stabilized gold nanospheres were found to be more toxic than gold nanostars. In that work, a star-looking shape was obtained using HEPES buffer as a reducing agent. On the other hand, Enea et al. reporting that nanospheres and nanostars capped with 11-mercaptoundecanoic acid (MUA) had similar cytotoxic properties [77]. In this case, the aspect of the shape fades into the background. However, in a work by Sultana et al. [78], flower-like gold nanoparticles and gold nanospheres with the same PEG shell and size were found to have different uptake and cytotoxic profiles. Authors associate the increased toxicity of gold nanoflowers with high surface roughness, mediating uptake and disturbance of the cytoskeleton.

Mesoporous silica nanoparticles (MSNs) have long been used for drug delivery and theranostics applications [79]. Little attention is usually paid to their shape since widespread protocols allow one to obtain mostly spherical particles. It was shown that silica nanorods with a length of about 450 nm (NLR450) are absorbed by cells much faster than rods with a length of 240 nm (NSR240) or spherical 100-nm particles (NS100) [80]. Consequently, NLR450 caused a greater cytotoxic effect, disturbances in the cytoskeleton structure and an increase in the number of apoptotic cells by 10% compared to other tested shapes. The authors also noted the influence of nanoparticles with various shapes on the migration and adhesion of cells, which may be associated with a cytoskeleton disruption.

Another widespread material, titanium dioxide, was also found to cause cytoskeleton damage in a shape-dependent manner [81]. Spherical TiO_2_ particles were discovered to penetrate the blood–brain barrier, probably through the disturbance of F-actin fibers. However, in addition to the shape, these particles also had a different crystalline phase.

When mentioning the crystalline phase, it should be noted that some authors equate the effect of the shape of particles and their crystalline phase [21]. It can be sometimes true; however, that the crystalline phase has more to do with surface morphology. For instance, TiO_2_ is known to have two crystalline phases—anatase and rutile—in addition to an amorphous form. Anatase was shown to have pronounced cytotoxic effects that were associated with the density of defect sites [82]. The anatase phase was also more toxic than rutile in a similar work, even though rutile particles were rod-like and sharp [83]. This observation suggests the necessity of the careful consideration of shape-dependent effects, especially in the context of inorganic particles.

Another example, in which a dependence of one kind can be mistaken for a dependence of another kind, is given in the work of Xu et al. [84]. Hydroxyapatite particles (HAP) with four different shapes were tested for general cytotoxicity manifestations (MTT assay), ALP activity, apoptosis assay (p53 and cytochrome C expression), and ROS generation. Almost all effects were in close correlation with particle-specific surface area but not with shape.

Usually, the shape of nanoparticles is discussed in the context of inorganic particles since a large number of approaches have been developed for them to create spatially anisotropic shapes. However, organic particles also have variations in shapes other than spherical. Biocompatible PLGA and PEG are some of the most common polymers used to create particles. However, it was shown that changing the shape of a particle from spherical to elongated leads to a significant increase in the particles’ cytotoxicity and induction of apoptosis [85]. The primary mechanism is assumed to be the rupture of lysosomes, followed by the launch of a cascade through caspase-3 and DNA damage.

Another example of organic particles with shape-dependent cytotoxicity is poly(3,4-ethylenedioxythiophene) (PEDT) polymer [86]. Oh et al. compared three types of PEDT nanomaterials with an average diameter of 55 nm. PEDT-1 was sphere-like, PEDT-2–rod-like, and PEDT-3 took the form of 1350 nm-long tubes. It was found that all samples caused a dose-dependent increase in LDH release in normal cells (IMR90 and J774A.1) by 5–50%. The ATP amount decreased in the same manner by 5–70%. The highest effect was observed for the smallest sphere-like particles. Furthermore, they caused a significant increase in the number of apoptotic cells (by 30%). However, the longest particles lead to considerable elevation of IL-1, IL-6, and TNF-α levels after 24 h of incubation.

Caspase-3 levels were elevated upon incubation of PC12 cells with graphene nanostructures of different dimensions [87]. The authors compared graphene layers (G) and carbon nanotubes (SWCNT), showing that SWNCT leads to LDH leakage, generation of ROS, and subsequent induction of apoptosis. The sharp shape of SWCNT was listed as the main reason for increased cytotoxicity.

Aluminum salts have long been widely used as an adjuvant for vaccines, whereas aluminum oxide (alumina) was used as an adsorbent. At the same time, methods for the synthesis of alumina usually include hydrothermal and ultrasonic-assisted approaches, in which precise control of the shape is not always possible [88]. Dong et al. managed to produce two types of alumina nanoparticles in the shapes of flakes and rods and studied them in detail using a metabolomics approach [89]. The main focus of the work was the study of brain cells, including primary cultures, which is especially important in the context of the medical use of alumina. Rod-like particles have been shown to have greater uptake ability, connected with increased cytotoxicity in astrocytes. Apoptotic markers and pro-inflammatory cytokine levels were increased in a dose-dependent and shape-dependent manner (similarly to the cytotoxic results). A careful analysis of 66 metabolites showed that nanorods cause more significant metabolic changes (21 unique differential metabolites (DMs) vs. 15 for nanorods and nanoflakes, respectively). These DMs mainly included amino acids, lipids, and carbohydrates.

##### Necrosis

In some cases, the process of cell death upon interaction with nanomaterials can follow the necrosis pathway [90]. From the therapeutic point of view, such an outcome is unfavorable, leading to inflammation [91]. However, in the context of screening potential nanomaterials for biomedical applications, it is imperative to conduct additional research for signs of necrotic cell death.

One of the most frequently used approaches to distinguish apoptotic and necrotic pathways is staining with FITC-labeled annexin. Using this technique, Huang et al. made a distinction between four shapes of hydroxyapatite nanoparticles (HAP) [3]. The authors found that cytotoxicity (and, similarly, the percentage of necrotic cells) rose in the order of plate > sphere > needle > rod. The difference between HAP plates and rods in the number of necrotic cells was almost twofold (17.13% vs. 9.67%, respectively). The impact of HAP on mitochondrial membrane potential and lysosome integrity followed the same shape-dependent trend. The authors concluded that the primary influence on cytotoxicity was exerted by the surface area, conductivity, and zeta potential of nanoparticles. However, cellular uptake of HAP was higher for the most toxic shapes—a fact that should be kept in mind.

The aggregation state of nanomaterials plays an essential role in cytotoxicity [92]. Lee et al. [93] showed that iron oxide rods that remain stable under physiological condition were more toxic than aggregated nanospheres. The number of necrotic cells was increased after incubation with rod-shaped particles by about 20% at a 200 μg/mL concentration. However, considering important observations connected with ROS generation and aggregation state, the straight correlation between shape and cytotoxicity is not clear.

##### Ferroptosis

In discussing the possible effect of nanoparticles on cell viability, the phenomenon of ferroptosis cannot be ignored. This form of cell death exists in many cancer cells and therefore attracts particular interest in cancer treatment [94]. The main object of studies in nanotechnology has become iron-containing nanoparticles, since it is iron that participates in the Fenton reaction, leading to ferroptosis. The primary mechanism involves the release of iron ions Fe^2+^ or Fe^3+^ after the nanoparticles enter the lysosomes or an acidic environment. The ion release process is directly related to the surface area of the nanoparticles, which is sometimes associated with the shape. Smaller particle sizes are preferable since they tend to generate more iron ions. Ferumoxytol is used as a contrast for MRI [95]. It is a stabilized magnetite (Fe_3_O_4_) nanoparticles with a size of about 50 nm [96]. This size range is greatly suited for cellular uptake. It has been shown that ferumoxytol can cause the polarization of macrophages in cancerous tumors, reducing their growth [97]. Tumor size was halved on day 21 in the ferumoxytol-treated group. To note, the polarization of macrophages plays an essential role in the treatment and diagnosis of cancer, which was the subject of our recent review [97]. Amorphous forms of iron also cause Fe^2+^ release and can be used to trigger the Fenton reaction [98]. The advantage of the amorphous form over the crystalline one was that, under the action of an acidic environment, the release of ions from the amorphous form was significantly higher. In the first 6 h, the release from the amorphous form was 100%, whereas for the crystalline form, it was only 20%.

Combined strategies, such as drug loading, are successfully implemented along with iron-oxide nanoparticles [99]. The delivery of hydrogen peroxide can be considered one of the relevant strategies [100]. A complex structure, consisting of a PLGA shell incorporated with magnetite nanoparticles and loaded with H_2_O_2_, was developed. The release of H_2_O_2_ was carried out using ultrasound. Tumor size was reduced by eight times compared with the control group on the 22nd day of the experiment.

Non-iron-containing engineered nanomaterials have also been reported to cause ferroptosis. A relatively large amount of papers have been published recently on manganese oxide (MnO_2_) particles [101,102,103]. Though no shape-dependent activity has been reported, the investigated particles possessed developed surfaces (nanoflowers or nanobubbles). Another example is two-dimensional transition metal dichalcogenides. WS_2_ and MoS_2_ nanosheets were shown to cause ferroptosis in epithelial (BEAS-2B) and macrophage (THP-1) cells [104]. However, cell death was connected with surface vacancy, not the 2D nature of the nanomaterials.

#### 4.1.2. Disturbance of Cell Function

Alterations of cell proliferation are not always accompanied by cell death and should be considered separately. In this section, we describe and discuss the shape-assisted alteration of cellular metabolism, genotoxicity, mammalian cell differentiation, as well as the influence of external stimuli on these processes (Figure 3).

##### Metabolic Alterations

In a typical investigation, various assays are used to assess cell proliferation by measuring cell metabolic activity. In most cases, they do not answer the question of what the exact cause of metabolic alterations is. Nevertheless, there are clear connections between nanoparticle shape and metabolic changes.

Six types of gold nanoparticle shapes were tested using the WST-1 assay on normal and cancer cell lines [5]. Aside from shape differences, spheres were the smallest ones (10 nm). Rods were about 40 nm in length; the rest of the particles exceeded 100 nm. Moreover, rods, prisms, and stars were synthesized in the presence of CTAB. A size-dependent effect was definitely found for rods and spheres, which were the most toxic ones. Shape-dependent behavior was obviously inherent to spiky nanoprisms and nanostars but not nanoflowers, the edges of which were smoother. Additionally, HeLa cells were much more resistant to the toxic effects of nanoparticles than HEK293T cells.

In the study by Arnida et al., several spherical- and rod-like nanoparticles were tested on prostate cancer cell line PC-3 [105]. It was found that nanospheres with 30–50 nm size have the highest cell uptake. PEGylation of particles drastically reduces cell uptake. No specific effects were associated with nanoparticle shape.

Research has shown that multiwalled carbon nanotubes (MWCNTs) have a similar structure to asbestos [106]. The latter is known for its ability to cause multiple pleural diseases [107]. Xu et al. [108] proved a hypothesis that the shape of MWCNT plays an important role in lung-associated diseases. Large (rod-like) and short (cotton candy-like) MWCNTs were administered in the rat lungs for 24 weeks. The authors found a shape-dependent formation of foci of mesothelial cell proliferation. Rod-like tubes also contributed to increased proliferating cell nuclear antigen (CD68) and pro-inflammatory cytokine levels.

Rod-shaped cerium oxide nanoparticles caused the same effect on pro-inflammatory cytokine levels [109]. The authors compared three sizes of rod-shaped particles (5.5–8.1 nm in diameter and 44–70 nm in length) and two sizes of octahedral-like particles (11.3–16.2 nm). All particles induced concentration-dependent cytotoxicity. All rod samples generate almost twice-higher TNF-α levels compared to octahedral particles. Additionally, the smaller the rods were, the more toxic they were. A similar dependence was found for LDH release. The smallest rod-like particles have the biggest surface area, which may be the reason for higher toxicity due to the release of ions. However, there was no ROS generation detected for all types of particles.

##### Genotoxicity

Nanoparticles are known to generate reactive oxygen species upon lysosomal or cytoplasmic degradation [110,111,112]. ROS is one of the primary causes of DNA damage [113]. However, ROS production can depend not only on chemistry, surface area, and size, but also on nanoparticle shape.

Rod-shaped zinc oxide nanoparticles were found to be significantly more genotoxic than sphere-like nanoparticles, mostly due to elevated ROS generation [114]. The authors assumed that increased ROS generation may be associated with a bigger surface area or higher uptake of rod-shaped particles, although no proof was provided.

Mesoporous silica nanoparticles cause genotoxicity in the DT40 cell line in a similar shape-dependent manner [115]. Rod-shaped particles were synthesized in the presence of CTAB. However, they were further calcined at 500 °C. In the test for chromosomal aberrations, rod-like particles showed a ~20% increase.

##### Cell Differentiation

Stem cells are of particular interest in many areas of biomedical research. Getting stem cells to turn into the required cells is challenging. It was found that some nanoparticles can force stem cells to differentiate, although the detailed mechanisms for this are not always known [1]. From the nanotechnology point of view, different nanoparticle shapes can lead to different effects.

Human mesenchymal stem cells (HMSCs) are multipotent cells that can replicate as undifferentiated cells. They have the potential to differentiate into mesenchymal tissues such as bones, muscle, marrow stroma, etc. [116]. Differentiation into bones and cartilage is of particular interest because of the prevalence of bone fractures, implanting of metal grafts, and other musculoskeletal system-associated disorders.

The effect of gold nanoparticles on osteogenic differentiation has been known for more than ten years [117,118,119]. However, the shape-dependent effect has not been studied well in detail. Li et al. studied full spectra of gold nanoparticles with three types of shapes (spheres, rods, and stars) and three different sizes [120]. The careful selection of particle parameters allowed the authors to distinguish shape effects clearly. Spheres and rods with dimensions close to 50 nm (but not stars) caused the most pronounced effect on the expression of osteogenic markers. Only nanorods with a length of 40 nm were found to be significantly cytotoxic, and therefore downregulated osteogenic markers.

Single- and multi-walled carbon nanotubes were also tested for their ability to alter HMSCs’ differentiation [121]. Both types inhibited cell proliferation, differentiation rate, and mineralization. The effect of single-walled CNTs was slightly more pronounced (within a 10% difference).

##### Ion Channel Disturbance

Though some researchers are not exploring the difference between particle shapes, shape dependence can still arise from external stimuli. For instance, when using an external magnetic field, the shape of the particles must be strictly defined. One of the uses of magnetic particles to control cell biological functions is related to the control of the activity of ion channels. Gregurec et al. [122] showed that magnetite nanodiscs with sizes of 98–226 nm could act as transducers of magnetic torque into cell stimuli [123]. Under the application of an external magnetic field with low flux and frequency, it could trigger Ca^2+^ influx.

A similar shape-based approach was described in the work of Lee et al. [124]. The authors assembled a complex structure, defined as an m-Torquer, from octahedron magnetite nanoparticles with predefined volume magnetization distributions. The m-Torquer was able to affect mechanosensitive ion channels both in vitro and in vivo.

##### Membrane Integrity

Although some particles can damage the cell membrane by themselves, sometimes their action can be controlled. In the case of plasmon resonant particles, their behaviour can be regulated by light excitation. The shape of some particles can serve as antennas for converting optical energy into heat [125]. Such a structure can allow the penetration of cell membranes in a controlled manner, which can be used for cell transfection. However, even slight alterations in nanoparticle morphology can lead to different responses to light irradiation [126].

In contrast to light-induced membrane penetration, Sun et al. used suspended magnetic hedgehog-like pollen grain particle for intracellular drug deliverys based on controlled membrane perforation under the action of a magnetic field [127]. Unlike the previous example, Qian et al. studied the interaction of the nickel urchin-like nanoparticle with tumor cells under the action of a rotating magnetic field [128]. Their results showed that the urchin-like particles induce a much higher necrosis rate than spherical nickel nanoparticles under the same rotating magnetic field. These examples revealed the potential of magnetic particles with branched structures for intracellular delivery or cell death action. However, these biological effects are closely interconnected and strictly depend on the size and shape of spikes on the particle’s surface and the magnetic field amplitude and frequency.

### 4.2. Microbial Cells

Microbial cells are one of the most important organisms on Earth. Their great diversity in genetic mechanisms, metabolic pathways, habitats in extreme environments, and survival strategies make them ideal organisms for biotechnology [129], molecular biology [130] and even in materials science [131]. Despite their laboratory and industrial applications [132], many diseases in the human body are connected with microbial cells. A comprehensive review describes not only the disease-mediated interaction of microbes and humans [133,134] but also emphasizes their crucial role in health. Hence, the control of microbial cells for humanity purposes may be helpful from several points: (i) the treatment of bacteria-mediated disease and (ii) microbial cells as factories for the production of valuable substances.

The treatment of bacterial infection and disease for almost one hundred years has been interconnected with antibiotics. Even though antibiotics are common and even traditionally used in therapy, researchers are now pushing back against an antimicrobial resistance wall [135]. The broad use and abuse of antibiotics has led to the formation of multidrug-resistant bacteria. The number of species resistant to traditional treatment is increasing rapidly and has become a major global healthcare problem. Over the last few years, the application of nanotechnology in biomedicine has become an alternative strategy for the treatment of disease, especially of microbial cell-mediated infection [136]. The nanoparticle platform has been used in drug delivery systems [10,137] and as part of implants coating against biofilms [138,139]. These examples expand the potential of synthetic and natural strategies to overcome antibiotic resistance and, consequently, serve as a universal approach for treating microbe-mediated disease.

The microbial cell factory is an alternative way to produce highly beneficial products such as pharmaceutics [140], chemicals [141], and biofuels [142]. The commonly used strategy to advance such systems is metabolic pathway design [141] and microbial cell engineering [143]. In contrast to treatment, the use of nanoparticles in microbial engineering is less widespread in the literature. Examples include protective nanoshells to overcome conditions in harsh environments [144,145], the integration of synthetic solar-to-chemical energy transformation pathways in non-photosynthetic organisms [146,147], and magnetic and light control of metabolic reactions [14,148]. The synergy of synthetic biology and materials science creates a unique possibility for the novel generation of microbes-based factories with remote control and the on-demand production of valuable products.

Despite the great potential of nanoparticles for biofilm therapy and improving microbial factories, most papers demonstrating the use of spherical nanoparticles focus on surface chemistry rather than the particle shape. To date, materials science allows particles to be designed for particular process-of-interest and creates a myriad of novel applications of nanoparticles in regulating microbes’ biochemistry, arising from nanoparticle shape. In this section, we outline the state of the art in the nanoparticle shape-dependent effect on microbial cells.

#### 4.2.1. Antimicrobial Shape-Dependent Effect

Nanoparticles of different shapes are widely used for the treatment of biofilms, as well as the induction of microbial cell death. Silver nanoparticles represent the majority of nanoparticle-mediated therapy. Ag NPs have received considerable attention due to their wide use as antimicrobial agents, which were shown to be very effective [149]. The toxicity of silver nanoparticles is usually associated with ROS generation, membrane damage, and ion release, which induce the dysregulation of DNA synthesis [148]. The pioneering work on the shape-dependent toxicity of silver nanoparticles was carried out by Pal et al. [150]. Triangular silver nanoplates with a {111} lattice plane have a stronger biocidal action than spherical and rod-shaped nanoparticles. The authors suggested the contribution of facets with distinct activity ({100} < {111}) for different shapes. Cheon et al. studied the shape-dependent toxicity of Ag nanoparticles with spherical-, disk- and plate-like shapes [151]. The antimicrobial effect was associated with different ion release kinetics displayed by nanoparticles with different shapes. The most pronounced antimicrobial activity was for spherical Ag NPs; the disk Ag NPs were more toxic than triangular plate Ag NPs. Acharya et al. studied the toxicity of spherical and rod-shaped silver particles [152]. They also provided data about the larger bactericidal effects of spheres due to their larger surface area, damaging the cell wall of both Gram-negative and Gram-positive bacteria. Thus, the shape of silver nanoparticles plays a crucial role in bactericidal activity with the dominance of the spherical shape above others.

The use of Au NPs has been documented for diverse applications [153] in biological systems due to their unique physio-chemical properties, which can be easily tuned by tuning the chemical routes [125]. Au NPs possess the ability to undergo localized surface plasmon resonance (SPR), which depends on the particle’s size, shape, and surface morphology. To date, the bactericidal action of gold nanoparticles as an antimicrobial drug carrier and as a photothermally-activated actuator has been shown [154]. Although size plays a crucial role [155] in triggering biological effects, it is necessary to consider the shape of the gold nanoparticles. Penders et al. studied the bactericidal performance of spheres, stars, and flowers with similar dimensions [156]. It is worth noting that particles’ antimicrobial activity was studied along with their biocompatibility to mammalian cells. The authors found that gold nanoflowers possessed the most promising non-cytotoxic mammalian cell behavior, with the greatest shape-dependent antibacterial activity-promising properties. Meanwhile, the nanospheres had no significant impact on *S. aureus* viability.

A comprehensive study of gold particle-mediated antimicrobial activity was carried out by Chmielewska and others [157]. The effect of Au rods, peanuts, and stars was assessed on a large set of Gram-positive and Gram-negative pathogenic bacteria. The study presented the well-pronounced bactericidal activity of a whole set of particles, primarily caused by ROS generation and related to the destruction of bacterial membranes. The gold peanuts have a prevailing bactericidal effect over other shapes. Hameed et al. demonstrated that gold nanocubes showed the most significant bactericidal properties, even at lower concentrations, against *E. coli*, *P. aeruginosa*, and *S. aureus*, followed by Au NSps (nanospheres) and Au NSts (nanostars) [158].

A special role in the literature is allocated for magnetically powered particles commonly based on Fe_3_O_4_ [159,160]. The magnetic field is a promising physical tool to control particle dynamic media due to the absence of energy dissipation in biological media, and the ability to generate heat by magnetic hyperthermia and mechanical force via magnetically induced particle oscillation and rotation under the action of an alternating magnetic field [161,162]. However, most papers describe the application of isotropic nanosized particles [163], as well as the mutual interaction of magnetic nanoparticle results in terms of vortex formation, which is efficiently operated by a magnetic field, and have been focused on accurate therapy [164,165,166] and cargo delivery [167,168]. Dong et al. designed a magnetic vortex (microswarm) consisting of porous magnetite particles, which, under the action of the rotating magnetic field, could disrupt the biofilm and generate ROS [169]. Moreover, the presented microswarm can be transported to confinement and narrow spaces for the effective disruption of biofilms. A similar concept was demonstrated by Hwang et al. [170]. Isotropic magnetic nanoparticles with catalytic activity were implemented as the building blocks of a catalytic antimicrobial robot designed as a result of the aggregation of nanoparticle and 3D molds. The robot could completely remove the biofilm without regrowing and be driven to the difficult-to-reach surfaces and even into human teeth.

Another exciting application of magnetic particles is inducing the shape-transformation of liquid metals. Gallium-based alloys remain liquid at room temperature and have a unique set of characteristics and behaviors, which originate from their simultaneous metallic and liquid natures. This has prompted the emergence of applications in various fields in catalysis, physical and synthetic chemistry [18], and biomedicine [171]. Elbourne et al. studied the shape transformation of liquid metal nanoparticles under the action of the magnetic field and the application of the designed shapes for the elimination of biofilms [172]. The bactericidal action of synthesized magnetic liquid particles was achieved only under the action of rotating magnetic fields, resulting in completely removed biofilms after 90 min of magnetic field exposition. The authors continued their work and used particles to efficiently eliminate multi-species biofilms, characterized as more complex structures due to their pathogenicity and resistance to the treatment and environmental factors [173].

Another bactericidal strategy involves the use of nanostructured surfaces inspired by nature. Insects commonly fabricate surfaces with high-aspect-ratio elements to damage the bacterial cell membrane [174]. The bactericidal effect depends on various parameters like height, width, spacing, nature, and geometry of the nanostructured surface. The influence of these parameters is the main focus of experimental [175,176] and theoretical work [177,178,179] and review papers [4,180,181], in which scientist have been looking for the mechanism of bacteria destruction and what exactly causes of cell death. Moreover, these surfaces are used for bactericidal effects and as a surface for improved growth and differentiation of mammalian cells [176,182]. Precisely, the mechanism of bacteria cell death involves the stretching and rupture of the cell membrane between the pillars, and the pillars are not expected to substantially deflect upon contact with bacteria. These facts have inspired scientist to look towards novel bioinspired methods for fabricating such nanostructured materials and using them in biomedicine. Valiei et al. studied the interaction of zinc oxide nanopillars with bacteria, which caused cell death [183]. This is interesting, but the interaction itself does not provide a bactericidal effect. The primary cause of cell death is the moving of the air–liquid interface between the media and the nanopillars. Linklater et al. fabricated vertically aligned carbon nanotubes [184] and demonstrated their significant antibacterial properties due to the storage and release of mechanical energy sufficient for bacteria membrane rupture. Taken together, the presented strategy would be helpful for the design of the next generation of implants to avoid the common problems with biofilms and induce the proliferation of mammalian cells.

#### 4.2.2. Shape-Dependent Metabolic Alterations

Microalgae represent one of the most promising organisms for the production of valuable metabolites such as biofuel and novel plant-based drugs [185,186]. However, large-scale cultivation remains the main obstacle in the development of economically beneficial techniques [185]. To date, nanomaterials represent an alternative way to increase the efficiency of algae cultivation. Eroglu et al. synthesized plasmonic nanostructures to enhance the accumulation of microalgal pigment [187]. They found that gold nanorods with maximum surface plasmon resonance at 710 nm had the most significant effect on combined pigment content up to 150 μmol m^−2^ s^−1^. SPR tuning by nanoparticle shape allows their introduction as optical filters, which could alter the production yield.

Cruces et al. studied the toxicity mechanism of the graphene oxide and multi-walled carbon nanotubes in *Microcystic aeruginosa* [188]. Despite the apparent mismatch in nanomaterial physicochemical properties, the author revealed similar dependencies in the inhibition of photosynthesis esterase activity and growth rate. The effects of metabolism alternation after nanomaterial exposition are mainly associated with interaction-induced alteration of photosynthetic electron transfer. Samei et al. studied the shape- and size-dependent effects of zinc oxide nanoparticles on freshwater microalgae [189]. Spherical ZnO NPs acted more destructively (50% cell death) to microalgal cells than nanorods (30% cell death) with the same concentration.

Methane is considered one of the most widely used and important biofuels [190]. Metallic and metal-oxide nanoparticles possess great potential to improve biomethane production [191]. Ambuchi et al. studied the combination of magnetic nanoparticles with multi-walled carbon nanotubes on biogas and methane generation from sludge with microorganisms, revealing improved biomethanogenesis upon nanomaterial treatment [192]. Some authors claimed that a similar effect of nanomaterials on methanogenesis enhancement was achieved by direct interspecies electron transfer due to transition metal oxide catalyst [191,192,193,194,195]. From this point, the shape tunability allows the control of methane generation in sludge or a microorganism consortium.

Respiration is a crucial metabolic activity, from which almost all energy is derived. Instead of eukaryotic cells, in prokaryotes this is carried out in cell membranes. Thus, the strategy to modulate the bacterial respiration chain does not involve the uptake of nanomaterials. The interaction between nanomaterials and bacteria, which has a high aspect ratio (tubular and rod). requiring extensive contact with bacterial membranes, has been shown to regulate metabolic processes in bacteria, even on the gene expression level [196,197].

Mortimer et al. studied the carbon nanomaterials and nanoceria influences on *Bradyrhizobium diazoefficiens*, the essential microorganism for nitrogen fixation in agriculturally beneficial plants [196]. They showed that nanomaterials with similar physical properties significantly altered the expression of genes associated with the nodulation competitiveness of *B. diazoefficien*. Moreover, they also suppressed root exudate-induced gene expression in bacteria, suggesting that nanomaterials may interfere with plant-microbe signaling. The same group published a paper about the influence of multi-walled carbon nanotubes, inducing differential regulation of 111 genes in *P. aeruginosa*. Interestingly, graphene, boron nitride, and carbon black caused differential regulation only of 44, 26, and 25 genes, respectively, despite reports of the common interaction mechanism of these materials with bacteria [197]. The upregulation of genes by carbon nanotubes involves a list of metabolic pathways, such as nitrogen metabolism, sulfur metabolism, membrane proteins, and a two-component regulatory system associated with antibiotic resistance.

Li et al. [198] found that spherical and nanosheet carbon nanomaterials positively affect the upregulation of genes involved in amino acid and carbohydrate metabolism. Zhang et al. studied the effect of chronic exposure to low concentrations of graphene oxide on *Escherichia coli* [199]. The interaction between graphene oxide and bacteria causes the significant activation of the Cpx envelope stress response, resulting in a more than two-fold increase of extracellular protease release and biofilm formation. Moreover, bacteria after graphene oxide exposition exhibited higher pathogenicity than control cells.

## 5. Conclusions and Future Prospects

The growing field of study regarding anisotropic nanoparticle shape originally arose from the tuning of particles’ optical, electrical, and magnetic properties. The further assessment of the interactions of these nanoparticles with biological objects opened up an exciting field of biomedical research. This review presents an overview of the most relevant reports for this broad field of biomedical and biotechnological topics. Nevertheless, the nanoparticles shape-dependent effect on cells has a huge potential in the control and up- and downregulation of biochemical processes, as well as whole-organism behavior.

The study of nanoparticle shape within the microbial context is usually focused on antimicrobial action with common mechanisms, such as ROS generation, ion release, and physical disruption of the membrane. The great challenge in the generation of new bioapplications is the material-guided control of metabolic activity of the microbes, horizontal gene transfer, and modulation of quorum sensing without a significant effect on cell viability. Within this field, future research might include the intracellular delivery of the reaction substrate, gene circuit, signaling molecules to achieve increases in metabolic performance and on-demand control of the biocatalysis, with the separate growth of microbes and synthesis of the valuable product phases. The manipulation of biofilm growth through the precise delivery of the gene construction and quorum sensing moieties opens an avenue of application in the biomedical treatment of pathogenic bacteria via biological routes and advanced bioreactors for high-yield production. Mechanobiology and intracellular delivery represent a tremendous achievement for mammalian cells. Therefore, the transfer of these approaches to microbial cells could ultimately prove successful in next-generation therapy and biotechnology applications. Since shape and crystal engineering of inorganic particles has demonstrated sufficient flexibility for biological applications, novel applications could arise from shape-dependent modulation of the processes of interest. For example, photosynthesis and nitrogen fixation control in microalgae via anisotropic particles could enable the modulation of plant growth and development for agricultural purposes. In addition to shape-control itself, the unique physicochemical properties of these anisotropic particles allow them to bring physical stimuli, such as light and magnetic and electric fields, potentially enabling the remote modulation of biochemical processes via biophysical cues. Finally, the potential for nanoparticle shape-mediated increases in antimicrobial resistance, taking care of possible negative consequences using developed methods, is also worth mentioning. Interdisciplinary research is essential in order to take the next step toward the advanced future of biomedicine and biotechnology.

The study of the effects of nanoparticle shape within the mammalian cell context could be focused on such areas as cell reprogramming, cell differentiation, and more precise control over biological functions. It should be kept in mind that nanoparticles can be used in the form of surface coatings, not just as stand-alone nanoparticles. Surface structuring with nanoparticles can provide great opportunities in areas of cell differentiation and reprogramming. An outstanding review on high-aspect-ratio nanostructures showed the crucial importance of interactions between cells and nanomaterials [180]. Surface structuring can be implemented for graft coatings. Doctors have long used various implants for vascular surgery as substitutes for vessels, bone prostheses, etc. The issue of the survival rate of these artificial objects is extremely important. The development of surface treatments may enable significant progress in the creation of highly biocompatible implants. Currently, work in this direction is actively promoted only in the field of bone surgery. However, for instance, proper graft surface functionalization for vessel substitution applications could prevent thrombocyte aggregation or the overgrowth of endothelial cells.

Cell reprogramming is the newest technology that can be used in regenerative medicine or in the treatment of genetic diseases. It is known that many physical parameters such as surface topology, stiffness, and charge can influence the reprogramming process [200]. It has already been shown that a specific surface microstructure is extremely important for cell reprogramming [201]. At the same time, there have been no works devoted to the detailed study or, most importantly, the comparison of various shapes of nanoparticles for cell reprogramming.

Ferroptosis is another example of an area where shape dependence has not been investigated. The anisotropic surface of nanoparticles can positively influence both the release of ions and lead to other effects, providing a synergistic effect on cancer cells.

Another common problem for nanobiomedicine-related sciences is the weak link between chemists and biologists. It is necessary to distinguish clearly the effects of particle shape from the functionalization of the surface, charge, size, and other properties. Occasionally, non-obvious properties such as charge density [202], crystallinity [203], aggregation state [99], and the structure of local inhomogeneities [204], are essential. Thus, it is necessary to optimize the synthesis procedures and create particles which differ only in their shape.

## Figures and Tables

**Figure 1 ijms-22-05266-f001:**
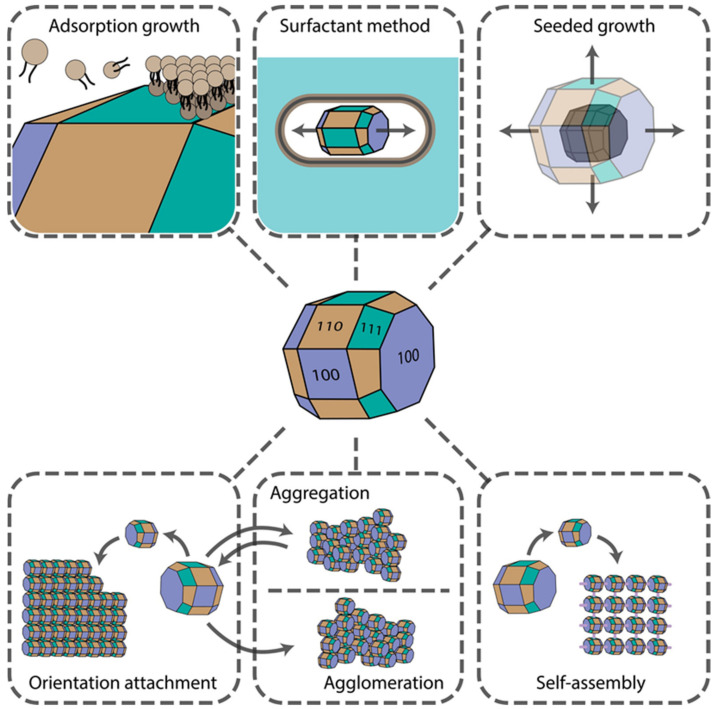
Main mechanisms underlying nanomaterial shape control.

**Figure 2 ijms-22-05266-f002:**
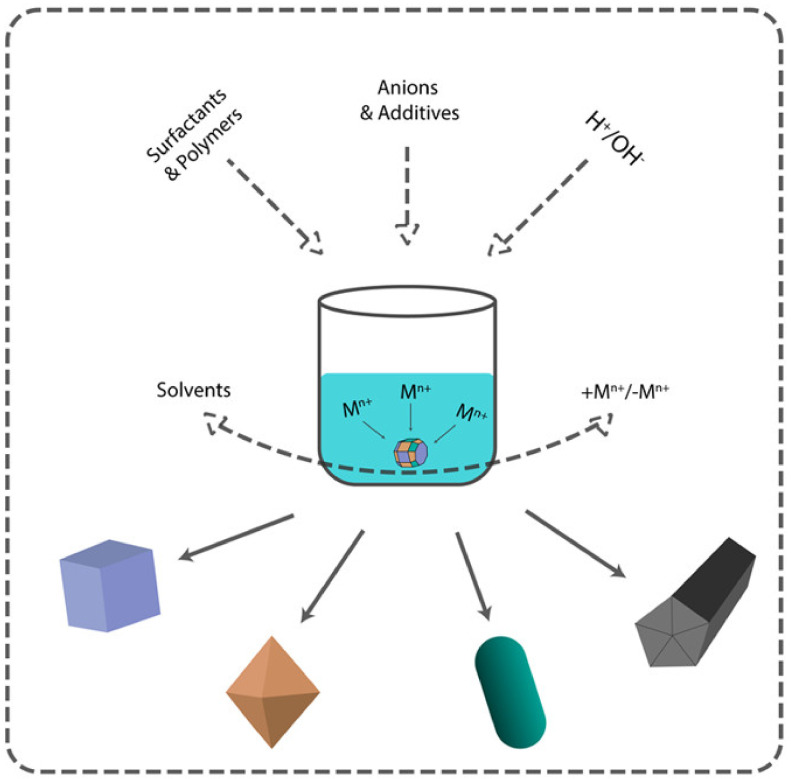
Main approaches for nanomaterial shape control.

**Figure 3 ijms-22-05266-f003:**
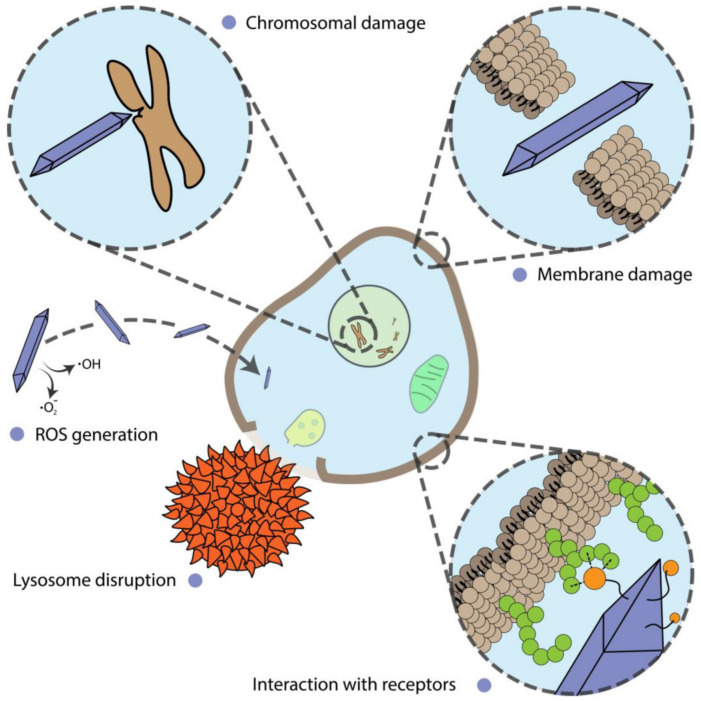
Different mechanisms of nanoparticle action on cells.

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
