# Peer review of "Nanomaterial Shape Influence on Cell Behavior"

_ijms, 2021, doi:10.3390/ijms22105266_

Round 1

Reviewer 1 Report

The paper has a good scientific soundness, since it gives a novel overview on this topic.

The accuracy of the division in section and subsection is high.

The description of phenomena using figures is well defined.

The use of English is surely outstanding and, also for this reason, the paper deserves to be published after very few changes.

I suggest adding an abbreviation list to this paper, according to the journal guidelines.

Among Line 511 and Line 512: there is an image without the proper caption.

I think that Paragraph among lines 807 and 813 should be moved to another section; instead, there should be the description of future perspectives.

Thank you.

Author Response

  • The paper has a good scientific soundness, since it gives a novel overview on this topic.
  • The accuracy of the division in section and subsection is high.
  • The description of phenomena using figures is well defined.
  • The use of English is surely outstanding and, also for this reason, the paper deserves to be published after very few changes.

Thank you for the appreciation of our work.

  • I suggest adding an abbreviation list to this paper, according to the journal guidelines.

We added an abbreviation list on Pages 19-20.

  • Among Line 511 and Line 512: there is an image without the proper caption.

The manuscript is now properly organized, and all caption are in place. 

  • I think that Paragraph among lines 807 and 813 should be moved to another section; instead, there should be the description of future perspectives.

We rewrite this part of the conclusions to deliver our thoughts on future prospects better. 

Reviewer 2 Report

In this manuscript, Kladko and colleagues systematically reviewed advances in modulating nanoparticle shape to control cellular and microbial behaviors. The authors covered various methods of synthesizing and controlling shape-specific nanomaterials. The shape dependence of nanoparticle-cell and nanoparticle-microbe interaction was then analyzed in detail. The review is well structured and well written. The figures are elegantly prepared to deliver clear outlines. While this is a great review manuscript, there are still some minor questions that need to be addressed (see comments below). These issues necessitate a minor revision to this manuscript before it can be considered for acceptance.

Additional Comments:

  1. It should be noted that additional cytotoxic pathways, such as ferroptosis, could be triggered by nanoparticles. Please include some examples of nanoparticles (of different sizes and shapes, if relevant) triggering ferroptosis.
  2. The fonts in Figure 3 is not consistent with Figures 1 and 2. Please modify to maintain consistency.
  3. The subsection “external stimuli” does not fit well into the “Disturbance of the cell function” section. This section should strictly focus on discussing how cell functions are affected by nanoparticle shapes. Functions such as “ion flux” or “cell membrane penetration” should be discussed rather than “external stimuli”.

Author Response

  • It should be noted that additional cytotoxic pathways, such as ferroptosis, could be triggered by nanoparticles. Please include some examples of nanoparticles (of different sizes and shapes, if relevant) triggering ferroptosis.

Thank you for this suggestion. We added the Ferroptosis subsection on Pages 11-12 and also some prospects on Page 19.

  • The fonts in Figure 3 is not consistent with Figures 1 and 2. Please modify to maintain consistency.

Figure 3 was modified accordingly.

  • The subsection “external stimuli” does not fit well into the “Disturbance of the cell function” section. This section should strictly focus on discussing how cell functions are affected by nanoparticle shapes. Functions such as “ion flux” or “cell membrane penetration” should be discussed rather than “external stimuli”.

We split and renamed this subsection and added some additional references. We hope it is now more contextual. The new subsections can be found on Page 14.

Reviewer 3 Report

The authors present a very complete review on the importance of nanomaterials' shape in biotechnology. The authors focus this review on two interesting perspectives, chemical and biological. I believe this article will be well received by both chemists and biologists. I find this review well written and well referenced.  For this reason, I recommend this review for publication without further revision.

Author Response

Thank you for your appreciation of our work!